# Why are so many pregnancies still unintended in Ghana? A closer look at factors influencing reproductive autonomy

Samuel Salu[1,2]*, Doreen Selasie Tay[1], Clinton Sekyere Frempong[2,3],
Dennis Kweku Mawuli Okyere[2,3], Prince Tsekpetse[4]

1 Department of Epidemiology and Biostatistics, Fred N. Binka School of Public Health, University of Health and Allied Sciences, Hohoe, Ghana, 2 Penuel-Charis Consultancy Limited, Hohoe, Ghana, 3 Department of Population and Behavioural Sciences, Fred N. Binka School of Public Health, University of Health and Allied Sciences, Hohoe, Ghana, 4 Department of Community and Public Health, Busitema University, Mbale, Uganda

* samuelsalu60@gmail.com

## Abstract

Unintended pregnancy remains a major public health challenge globally, with particularly high prevalence in sub-Saharan Africa. In Ghana, despite improvements in reproductive health service delivery, unintended pregnancies persist. This study examined the prevalence and factors influencing unintended pregnancy among sexually active women in Ghana. The study used data from the 2017 Ghana Maternal Health Survey (GMHS), focusing on 1,453 sexually active women aged 15–49 years. Descriptive statistics, chi-square tests, and logistic regression models were employed to assess the prevalence and factors associated with unintended pregnancy. The prevalence of unintended pregnancy among participants was reported to be 57.9%. Multivariable analysis revealed that women aged 20–39 years were significantly more likely to experience unintended pregnancy compared to those aged 15–19 years. Women with primary education were less likely to report unintended pregnancies as compared to those with no education (aOR = 0.59, 95% CI: 0.42–0.82). Living with a partner (aOR = 1.68, 95% CI: 1.27–2.23) and residing in the Savannah region (aOR = 5.52, 95% CI: 3.93–7.76) were associated with higher odds of unintended pregnancy. Conversely, internet use (aOR = 0.48, 95% CI: 0.30–0.66) showed protective effects against unintended pregnancies. The study concludes that unintended pregnancy was highly prevalent among sexually active women in Ghana. The findings highlight that public health interventions should be specifically targeted towards women above 20 years, those living with partners, and residents in the Savannah region. The findings also revealed that targeted interventions could leverage the protective effects of internet use by expanding access to reproductive health information via digital platforms and improving educational attainment to reduce the burden of unintended pregnancies among sexually active women.

**Data availability statement:** Complete dataset can be requested from the DHS program on reasonable request. The IRB-approval procedures for DHS public-use datasets do not allow in anyway the respondents, households or sample communities to be identified. To have access to the data, a registered request of a research project must be submitted and approved by the DHS. The instruction for requesting for the DHS data can be found on their website https://dhsprogram.com/data/Access-Instructions.cfm.

**Funding:** The author(s) received no specific funding for this work.

**Competing interests:** The authors have declared that no competing interests exist.

## Introduction

Unintended pregnancy refers to a pregnancy that is mistimed, unplanned, or unwanted at the time of conception and remains a major public health concern worldwide [1]. It often arises from non-use, inconsistent use, or incorrect use of effective family planning methods [2]. According to the United Nations Sexual and Reproductive Health Agency, nearly half of all pregnancies globally are unintended, amounting to an estimated 121 million unintended pregnancies annually between 2015 and 2019, which is approximately 331,000 per day [3]. Although there has been a global decline in unintended pregnancies over the past few decades, many low- and middle-income countries (LMICs) still experience a disproportionately high burden [3,4]. For instance, while the rate of unintended pregnancies is approximately 35 per 1,000 women aged 15–49 in Europe and North America, it is significantly higher in less developed regions [1].

In sub-Saharan Africa (SSA), the issue is even more alarming. The regional average is 91 unintended pregnancies per 1,000 women aged 15–49 in low-income regions, and more than double the rate in high-income regions [1]. Furthermore, there is substantial variation within SSA, with reported prevalence ranging from 10.8% in Nigeria to 54.5% in Namibia [3,5]. An estimated 14 million unintended pregnancies occur each year among sexually active women in SSA, largely due to structural and systemic barriers to reproductive healthcare [3]. The consequences of these pregnancies are far-reaching as they increase the risk of unsafe abortions, maternal mortality, malnutrition, mental health issues, school dropout, and vertical transmission of HIV [1,6]. It is estimated that 1 in 16 women in SSA experience psychosocial and economic burdens stemming from unintended pregnancies [4]. These issues are often exacerbated among adolescents and young women aged 15–24, who confront higher fertility rates, lower contraceptive knowledge, and limited access to family planning services [1,6].

In Ghana, the problem persists despite improvements in sexual and reproductive health service delivery. Studies in Ghana revealed that while urban women have greater access to family planning services, unintended pregnancies remain prevalent particularly in rural areas, where access to contraception and reproductive health education is often limited [7,8]. These findings reflect broader trends across SSA and highlight the impact of socioeconomic status, education level, place of residence, and cultural beliefs on reproductive health outcomes in Ghana. The persistence of unintended pregnancies in both rural and urban areas points to gaps in contraceptive access, inadequate reproductive health education, and the influence of socio-cultural norms [9,10]. The extant literature has identified important sociodemographic factors that influence the likelihood of unintended pregnancy, including education level, age, marital status, parity, residence (urban/rural), wealth index, occupation, healthcare decision-making autonomy, birth interval, exposure to family planning messages, knowledge and use of contraceptives, and experiences of sexual violence [1,2,4,5,7,8,11].

While these studies have explored the prevalence and determinants of unintended pregnancy, most of their analyses have been conducted among the general

reproductive aged women [4,5,8] without disaggregating those who are sexually active. This aggregation may produce biased or diluted estimates since women who have never engaged in sexual activity are biologically excluded from pregnancy risk. For instance, a study in Ethiopia by Jejaw et al., [5] found unintended pregnancy prevalence to be 31.66% among all reproductive-age women. Whereas, a study among sexually active women in Nigeria by Lawani et al., [12] reported prevalence of 43.8%. This suggests that the true burden among sexually active women may be substantially underestimated when data are pooled across all women.

Essentially, factors such as education level, age at marriage and sex education displayed stronger associations with unintended pregnancy when restricted to sexually active women [12] revealing the influence of relational dynamics that differ from the general reproductive-age population which found factors such as maternal occupation, household size, parity and women wealth to be associated with unintended pregnancy [5,13]. Such subgroup analyses can thus refine understanding of who is most at risk and why offering clearer entry points for reproductive health interventions.

The present study contributes new insights by focusing on sexually active women, as Ghana's Family Planning 2030 initiative emphasizes equity and precision in targeting subgroups most at risk of poor reproductive outcomes. By identifying predictors of unintended pregnancy among this population, program managers can more effectively allocate resources toward interventions that address specific barriers within this subgroup.

## Materials and methods

### Ethics statement

To perform these analyses, we sought formal approval from MEASURE DHS. The dataset can be accessed from the MEASURE DHS database at http://dhsprogram.com/data/available-datasets.cfm.

### Data source and design

This study utilized secondary data from the 2017 GMHS, a nationally representative survey conducted between June 15 and October 12, 2017. The Ghana Statistical Service (GSS) and the Ghana Health Service (GHS) implemented the survey with technical assistance from Inner City Fund (ICF) International under the Demographic and Health Surveys (DHS) Program. The survey employed a cross-sectional design to provide reliable estimates on key maternal and reproductive health indicators in Ghana [14].

The survey employed four types of questionnaires: the household questionnaire, the woman's questionnaire, the verbal autopsy questionnaire, and the health facility questionnaire. This study specifically used data collected through the women's questionnaire, which targeted women aged 15–49 years [14]. Eligible women were asked about various maternal health issues, including pregnancy history, pregnancy outcomes (live birth, miscarriage, stillbirth, and abortion), and timing of all pregnancies. Only women who provided informed consent were interviewed [14]. The survey collected data at a single point in time, allowing for the analysis of maternal and reproductive health indicators among women of reproductive age (15–49 years) [14].

### Study population

This study's analysis was restricted to sexually active women aged 15–49 years, defined in the GMHS as women who reported *"ever had sex"*. Respondents who indicated they had never had sex were excluded because they were biologically ineligible for pregnancy. This definition provides a more accurate estimate of unintended pregnancy by focusing on women actually exposed to the risk of conception. Although this definition does not specify a fixed recall period, it is appropriate for capturing lifetime sexual exposure in population-based surveys where many women may have intermittent sexual activity.

 

## Sampling strategy and sample size

The GMHS employed a stratified two-stage cluster sampling design based on the 2010 Population and Housing Census. In the first stage, 900 enumeration areas were selected with probability proportional to size, with 466 from urban areas and 434 from rural areas, stratified by region and residence type [14]. In the second stage, 30 households were systematically selected from each enumeration area, resulting in a total sample of 27,001 households. All women aged 15–49 years who were either permanent residents of the selected households or had spent the night before the survey in the household were eligible for interview [14]. Of the 25,304 eligible women identified, interviews were completed with 25,062 [14].

In preparing the data for analysis, exclusions were applied sequentially based on the survey skip pattern and data completeness. The first variable considered was *"age at first sexual intercourse"* followed by *"unintended pregnancy"*. An unweighted frequency distribution indicated that 3,665 women reported never having had sex, 5 refused to respond, and 21,392 reported ever having had sexual intercourse. After excluding women who reported never having had sex and those with missing data on the unintended pregnancy variable, the final analytical sample comprised 1,453 sexually active women who had complete information on pregnancy intention. These exclusions were necessary to maintain analytical accuracy but may introduce selection bias if excluded respondents differ systematically in sociodemographic characteristics. To minimize this potential bias, sample weights were applied and sociodemographic characteristics of the excluded group were examined descriptively to ensure broad similarity to the retained population (Fig 1).

## Variables

**Dependent variables.** The dependent variable in this study was *unintended pregnancy*. Unintended pregnancy was recorded in the dataset based on the question, *"When you became pregnant, did you intend to get pregnant at that time?"* Responses were coded as *"Yes" (1)* or *"No" (2)*. These responses were recoded into *1 "No"* and *0 "Yes.*

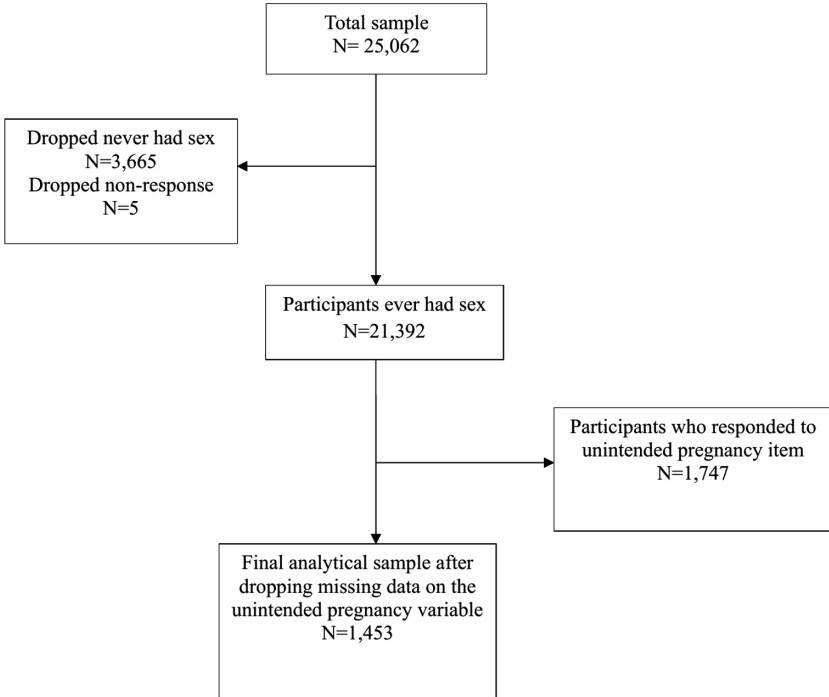

**Fig 1. Flow diagram of participants selection for the study.**

**Independent variables.** Based on a thorough review of relevant literature [1–5,8,10,11] and the availability of the variables in the dataset, 11 independent variables were selected for inclusion in this analysis. These variables were carefully recoded to suit the analytical objectives of the study. Maternal age was categorized into seven groups (15–19 years, 20–24 years, 25–29 years, 30–34 years, 35–39 years, 40–44 years, and 45–49 years). Age at first sexual intercourse was grouped as (15 years, 15–24 years, and 25 years and above). Educational attainment was classified into four levels (no education, primary, secondary, and tertiary). Place of residence was categorized as rural or urban. Region was grouped into three ecological zones (coastal, forest, and savannah). Living arrangement was assessed by whether respondents were currently living with a partner (yes or no). Access to media was measured using three variables (use of the internet in the last twelve months, listening to the radio, and watching television) all coded as yes or no. Parity was captured using a binary variable indicating whether the respondent had ever given birth (yes or no). Knowledge of the fertile period was also measured as (yes or no).

## Statistical analysis

All statistical analyses were performed using STATA software version 17.0 (StataCorp, College Station, TX, USA). The complex survey design was explicitly accounted for using the "**svyset**" commands in STATA to adjust for clustering, stratification, and sampling weights thereby ensuring population-level representativeness and accurate variance estimation. Descriptive statistics were performed to summarize the sociodemographic characteristics of the study population. To assess the relationship between sociodemographic factors and unintended pregnancy, Chi-square tests were performed for categorical variables. These tests provided estimates of the associations, including percentages and 95% confidence intervals (CIs).

For bivariate analysis, each independent variable was individually examined for its association with unintended pregnancy using logistic regression. The crude odds ratio (cOR), 95% CI, and p-values were evaluated for each variable. Variables with p-values ≤ 0.05 from the bivariate analysis were included in the multivariable logistic regression model to evaluate their independent effects on unintended pregnancy. Before running the multivariable model, the potential for multicollinearity among independent variables was assessed using variance inflation factors (VIF). The mean VIF was calculated as 1.44. No VIF values exceeded 5, indicating that multicollinearity was not a concern. The multivariable logistic regression model was used to calculate adjusted odds ratios (aOR), along with 95% CIs and p-values, to determine the strength of associations between socio-demographic factors and unintended pregnancy. Statistical significance was set at $p < 0.05$, and the results were interpreted in the context of their respective confidence intervals.

## Results

### Sociodemographic characteristics of participants

From Table 1 below, the majority of participants were aged between 25 and 29 years (26.4%). When assessing the age at first sexual intercourse, most respondents reported initiation between the ages of 15 and 24 (84.1%). In terms of educational attainment, over half of the participants had attained primary education (54.6%). A little over half of the respondents resided in rural areas (53.0%). Regionally, the majority came from the Coastal zone (43.6%). Living arrangements revealed that a greater proportion were living with a partner (67.4%) compared to those who were not (32.4%). Regarding media access and information, a large percentage had not used the internet in the past year (65.4%), though a significant majority listened to the radio (76.2%) and watched television (76.2%). Most respondents (77.8%) had ever given birth, and a substantial proportion (88.4%) demonstrated knowledge of their fertile period. Interestingly, a notable 57.9% of the participants reported experiencing unintended pregnancy.

**Table 1. Sociodemographic characteristics of participants.**

| Variable | Frequency (n = 1,453) | Percentage (%) |
|---|---|---|
| **Age** | | |
| 15 – 19 years | 129 | 8.9 |
| 20 – 24 years | 293 | 20.2 |
| 25 – 29 years | 383 | 26.4 |
| 30 – 34 years | 355 | 24.4 |
| 35 – 39 years | 218 | 15.0 |
| 40 – 44 years | 58 | 4.0 |
| 45 – 49 years | 18 | 1.2 |
| **Age at first intercourse** | | |
| Below 15 years | 198 | 13.6 |
| 15 – 24 years | 1222 | 84.1 |
| 25 years and above | 34 | 2.3 |
| **Highest educational level** | | |
| No education | 343 | 23.6 |
| Primary | 794 | 54.6 |
| Secondary | 201 | 13.8 |
| Tertiary | 116 | 8.0 |
| **Place of residence** | | |
| Rural | 770 | 53.0 |
| Urban | 683 | 47.0 |
| **Region** | | |
| Coastal | 633 | 43.6 |
| Forest | 551 | 37.9 |
| Savannah | 267 | 18.5 |
| **Living with partner** | | |
| No | 473 | 32.6 |
| Yes | 980 | 67.4 |
| **Used the internet last 12 months** | | |
| No | 951 | 65.4 |
| Yes | 502 | 34.6 |
| **Listen to radio** | | |
| No | 346 | 23.8 |
| Yes | 1107 | 76.2 |
| **Watches television** | | |
| No | 346 | 23.8 |
| Yes | 1107 | 76.2 |
| **Ever given birth** | | |
| No | 323 | 22.2 |
| Yes | 1130 | 77.8 |
| **Knowledge of fertile period** | | |
| No | 169 | 11.6 |
| Yes | 1284 | 88.4 |
| **Unintended pregnancy** | | |
| No | 611 | 42.1 |
| Yes | 842 | 57.9 |

## Sociodemographic characteristics and unintended pregnancy

Unintended pregnancy was most prevalent among women aged 25–29 years (64.3%) and 30–34 years (63.1%). With regard to age at first intercourse, unintended pregnancy was highest among those whose sexual debut occurred at age 25 and above (78.3%). Urban residents had a slightly higher proportion of unintended pregnancy (60.3%). Regionally, the Savannah region recorded the highest prevalence of unintended pregnancy, with 232 out of 267 women (86.3%, 95% CI: 83.0-89.0) reporting this prevalence. Participants living with a partner had a significantly higher prevalence (63.6%) of unintended pregnancy. Internet users (63.7%), those who did not listen to radio (62.2%), and those who had ever given birth (59.5%) reported higher unintended pregnancy compared to their counterparts (Table 2).

## Factors influencing unintended pregnancy

From Table 3, age remained a significant predictor of unintended pregnancy. Women aged 20–24 had over three times the odds (aOR = 3.02, 95% CI: 1.81–5.06) of reporting unintended pregnancy compared to those aged 15–19. The odds further increased for ages 25–29 (aOR = 3.85, 95% CI: 2.20–6.76), and 30–34 (aOR = 3.88, 95% CI: 2.13–7.07). The 35–39 group also had significantly higher odds (aOR = 3.18, 95% CI: 1.68–6.01), while older age groups did not show significant associations. Compared to women with no education, those with primary education were significantly less likely to report unintended pregnancy (aOR = 0.59, 95% CI: 0.42–0.82, p = 0.002). Women from the Savannah region also had significantly higher odds of experiencing unintended pregnancy (aOR = 5.52, 95% CI: 3.93–7.76, p < 0.001) compared to those from the Coastal region. Living with a partner was also significantly associated with higher odds (aOR = 1.68, 95% CI: 1.27–2.23, p < 0.001). Interestingly, internet use had a protective effect in the adjusted model, with users showing reduced odds of unintended pregnancy (aOR = 0.48, 95% CI: 0.30–0.66, p = 0.033). Knowledge of the fertile period also showed a slight protective association (aOR = 0.85, 95% CI: 0.65–0.98, p = 0.037). Other variables, including radio and television use, age at first intercourse, and whether the participant had ever given birth, did not retain statistical significance after adjustment.

## Discussion

This study sought to examine the prevalence and factors influencing unintended pregnancy among sexually active women in Ghana. Results from this study showed that the prevalence among this population is excessively high. Notably, more than half (57.9%) of the pregnancies in this study were reported to be unintended. This finding is higher when compared to previous studies in Ghana (34–45%) [15,16], Ethiopia (31.66%) [5], Nigeria (43.8%) [12], and Uganda (44.5%) [17]. The variations observed in findings could be because these studies were conducted among the general reproductive age women without restricting the analysis to only sexually active women. Such inclusion likely may have underestimated the true prevalence of unintended pregnancies, as sexually inactive women may also have been represented in those findings. Beyond these plausible methodological differences, the high prevalence observed in this study may also reflect Ghana's evolving reproductive health challenges. Persistently high rates of unmet need for family planning, inconsistent contraceptive use, and socio-cultural norms that discourage open discussions about sexuality continue to influence women's reproductive autonomy [18–21]. Additionally, structural challenges within the Ghana's health system, specifically the periodic stockouts of preferred contraceptive methods, variations in counselling quality, and widespread misconceptions about side effects may further contribute to the observed magnitude of unintended pregnancy [19,22].

The findings from this study revealed that age was significantly associated with unintended pregnancy. This was such that the prevalence among participants increased as age increases. Women aged from 20 to 39 years were more likely to have unplanned pregnancies. Previous study by Nyarko [23] also found that older women were 4.85 times more likely to have unplanned pregnancies compared to respondents aged 15–19. A plausible explanation could be that, older women are likely to be in romantic relationships, and/or engaged in sexual practices. In many Ghanaian communities,

**Table 2. Socio-demographic characteristics and unintended pregnancy.**

| Variable | Unintended Pregnancy % [95% CI] | | P-value |
| --- | --- | --- | --- |
| | **No** | **Yes** | |
| **Age** | | | |
| 15 – 19 years | 69.7 [59.4, 78.3] | 30.3 [21.7, 40.6] | **<0.001** |
| 20 – 24 years | 44.5 [37.3, 51.8] | 55.5 [48.2,62.7] | |
| 25 – 29 years | 35.7 [29.5, 42.4] | 64.3 [57.6, 70.5] | |
| 30 – 34 years | 36.9 [30.1, 44.3] | 63.1 [55.7, 69.9] | |
| 35 – 39 years | 37.7 [29.1, 47.2] | 62.3 [52.8, 70.9] | |
| 40 – 44 years | 55.3 [38.9, 70.6] | 44.7 [29.4, 61.1] | |
| 45 – 49 years | 52.0 [27.1, 75.9] | 48.0 [24.1, 72.9] | |
| **Age at first intercourse** | | | |
| Below 15 years | 41.6 [33.3, 50.5] | 58.4 [49.5, 66.7] | 0.143 |
| 15 – 24 years | 42.7 [38.7, 46.8] | 57.3 [53.2, 61.3] | |
| 25 years and above | 21.7 [10.1, 40.6] | 78.3 [59.4, 89.9] | |
| **Highest educational level** | | | |
| No education | 28.0 [22.3, 34.4] | 72.0 [65.6, 77.7] | **<0.001** |
| Primary | 51.3 [46.5, 56.1] | 48.7 [43.9, 53.5] | |
| Secondary | 40.6 [31.5, 50.4] | 59.4 [49.6, 68.5] | |
| Tertiary | 22.8 [13.9, 35.0] | 77.2 [65.0, 86.1] | |
| **Place of residence** | | | |
| Rural | 44.2 [39.4, 49.0] | 55.8 [51.0, 60.6] | 0.203 |
| Urban | 39.7 [34.8, 44.7] | 60.3 [55.3, 65.2] | |
| **Region** | | | |
| Coastal | 46.1 [40.3, 51.9] | 53.9 [48.1, 59.7] | **<0.001** |
| Forest | 51.3 [45.8, 56.8] | 48.7 [43.2, 54.2] | |
| Savannah | 13.7 [11.0, 17.0] | 86.3 [83.0, 89.0] | |
| **Living with partner** | | | |
| No | 53.8 [47.9, 59.5] | 46.2 [40.5, 52.1] | **<0.001** |
| Yes | 36.4 [32.4, 40.6] | 63.6 [59.4, 67.6] | |
| **Used the internet last 12 months** | | | |
| No | 45.1 [41.1, 49.2] | 54.9 [50.8, 58.9] | **<0.007** |
| Yes | 36.3 [31.0, 41.9] | 63.7 [58.1, 69.0] | |
| **Listen to radio** | | | |
| No | 37.8 [31.6, 44.5] | 62.2 [55.5, 68.4] | 3.345 |
| Yes | 43.4 [39.5, 47.3] | 56.6 [52.7, 60.5] | |
| **Watches television** | | | |
| No | 40.2 [33.6, 47.2] | 59.8 [52.8, 66.4] | |
| Yes | 42.6 [38.7, 46.7] | 57.4 [53.3, 61.3] | |
| **Ever given birth** | | | |
| No | 47.6 [40.6, 54.6] | 52.4 [45.4, 59.4] | |
| Yes | 40.5 [36.5, 44.6] | 59.5 [55.4, 63.5] | |
| **Knowledge of fertile period** | | | |
| No | 38.3 [28.9, 48.6] | 61.7 [51.4, 71.1] | 0.442 |
| Yes | 42.6 [38.8, 46.4] | 57.4 [53.6, 61.2] | |

**Table 3. Factors influencing unintended pregnancy.**

| Variable | cOR [95% CI], p-value | aOR [95% CI], p-value |
|---|---|---|
| **Age** | | |
| 15 – 19 years | Ref. | Ref. |
| 20 – 24 years | **2.90(1.90-4.43), <0.001** | **3.02(1.81-5.06), <0.001** |
| 25 – 29 years | **3.56(2.34-5.40), <0.001** | **3.85(2.20-6.76), <0.001** |
| 30 – 34 years | **3.38(2.21-5.17), <0.001** | **3.88(2.13-7.07), <0.001** |
| 35 – 39 years | **2.91(1.85-4.56), <0.001** | **3.18(1.68-6.01), <0.001** |
| 40 – 44 years | **2.09(1.11-3.90), 0.023** | 1.87(0.84-4.17), 0.124 |
| 45 – 49 years | 1.21(0.48-3.04), 0.691 | 1.06(0.30-3.70), 0.932 |
| **Age at first intercourse** | | |
| Below 15 years | Ref. | Ref. |
| 15-24 years | 1.20(0.87-11.65), 0.262 | 1.12(0.77-1.65), 0.548 |
| 25 years and above | 1.70(0.72-3.98), 0.225 | 1.90(0.72-4.96), 0.193 |
| **Highest educational level** | | |
| No education | **Ref.** | **Ref.** |
| Primary | **0.30(0.23-0.39), <0.001** | **0.59(0.42-0.82), 0.002** |
| Secondary | **0.56(0.38-0.82), 0.003** | 0.85(0.53-1.36), 0.507 |
| Tertiary | 1.18(0.69-2.03), 0.552 | 1.52(0.80-2.91), 0.201 |
| **Place of residence** | | |
| Rural | Ref. | Ref. |
| Urban | 0.97(0.78-1.21), 0.786 | 1.08(0.82-1.43), 0.571 |
| **Region** | | |
| Coastal | **Ref.** | **Ref.** |
| Forest | 0.98(0.74-1.30), 0.906 | 1.07(0.79-1.44), 0.667 |
| Savannah | **5.64(4.20-7.57), <0.001** | **5.52(3.93-7.76), <0.001** |
| **Living with partner** | | |
| No | **Ref.** | **Ref.** |
| Yes | **2.63(2.07-3.34), <0.001** | **1.68(1.27-2.23), <0.001** |
| **Used the internet last 12 months** | | |
| No | **Ref.** | **Ref.** |
| Yes | **1.45(1.14-1.83), 0.002** | **0.48(0.30-0.66), 0.033** |
| **Listen to radio** | | |
| No | **Ref.** | Ref. |
| Yes | **0.71(0.55-0.91), 0.006** | 0.86(0.64-1.17), 0.341 |
| **Watches television** | | |
| No | Ref. | Ref. |
| Yes | 0.83(0.65-1.05), 0.119 | 1.30(0.94-1.78), 0.109 |
| **Ever given birth** | | |
| No | Ref. | Ref. |
| Yes | 1.29(0.99-1.66), 0.055 | 0.81(0.55-1.19), 0.291 |
| **Knowledge of fertile period** | | |
| No | Ref. | Ref. |
| Yes | 0.79(0.57-1.09), 0.155 | 0.85(0.65-0.98), 0.037 |

childbearing during the twenties and thirties is socially expected, yet conversations about limiting or spacing births often occur within unequal power structures that reduce women's autonomy in deciding when to become pregnant [24]. These cultural expectations, combined with partner influence, concerns about side effects, and persistent misconceptions about contraception may contribute to inconsistent or non-use of modern contraceptives among this age group [25], thereby increasing the likelihood of unintended pregnancies. Contrasting findings include those of Ahinkorah et al., [26], who found that adolescents (15–19 years) have the highest rates of unintended pregnancy. This difference may be explained by the nature of the sample population used in this study, which focused on reproductive health decision-making capacity measured through indicators such as decision-making on sexual intercourse and condom use.

The current study also found that level of education influenced unintended pregnancy, such that women with primary education were less likely to report unintended pregnancy as compared to those with no formal education. Several studies have shown a higher risk of unintended pregnancy among populations with no formal education [26,27]. Particularly, a study conducted among sexually active women in Nigeria reported similar association [12]. This could be explained that uneducated women may have little or no knowledge about the existing family planning and safe sex practices, and hence are more vulnerable to becoming pregnant unintentionally [28]. A negative health implication could be the increase of adverse outcomes for both mothers and children, including delayed prenatal care, risk of maternal mortality and death, low birth weight and preterm delivery, and reduced likelihood of breastfeeding initiation. In Ghana, women without formal education often have limited access to accurate reproductive health information, restricted exposure to family planning messages, and lower confidence in navigating contraceptive options [29]. Despite national efforts from the Ghana Health Service family planning communication campaigns and the decades of school-based sexual and reproductive health education, women with no education frequently remain excluded from these platforms due to literacy barriers, poverty, and limited interactions with the formal health system [30].

Culturally, low educational attainment tends to intersect with gender norms that limit women's autonomy in sexual and reproductive decision-making. In many Ghanaian households, decisions regarding contraceptive use are influenced or controlled by male partners, and women with limited education may have fewer negotiation skills or less power to challenge partner preferences [31]. Consequently, they may be unable to consistently use modern contraceptives, increasing their vulnerability to unintended pregnancy.

We also found that women from the Savannah region were five times more likely to report unintended pregnancy. This association reflects the longstanding reproductive health disparities in northern Ghana, where geographic, socio-economic, and cultural factors intersect to limit women's ability to exercise reproductive autonomy. The Savannah Region is characterized by high levels of poverty, lower female literacy, and widespread dependence on subsistence livelihoods, which collectively reduce women's access to accurate reproductive health information and modern family planning services [32,33]. Health service availability is also uneven as many communities are hard-to-reach and sparsely populated relying heavily on CHPS compounds that often experience staffing gaps, limited method mix, and inconsistent contraceptive counselling [34]. These structural constraints significantly restrict women's access to reliable contraception and quality reproductive health services. This is in agreement with a study conducted in Bangladesh by Haque et al., [35], which also found that living in rural areas increases the risk of reporting unintended pregnancies.

Furthermore, the cultural norms in the northern part of Ghana may further shape reproductive behaviour. In many communities, decision-making around fertility and contraceptive use is strongly patriarchal, with husbands or male partners holding considerable influence over whether modern contraceptives are used, as well as the number of children they desire [36]. The high value placed on large families and the social prestige associated with childbearing can discourage women from seeking contraception or openly discussing fertility preferences. In contrast, a study conducted in Indonesia by Laksono et al., [37] discovered that women in rural locations have a lower chance of becoming pregnant unintentionally than those in urban areas. According to the study, living in an urban area raises the risk of unwanted pregnancies among Indonesian single women.

Moreover, women living with their partner were found to report more unintended pregnancies as compared to their counterparts who were living alone. This finding resonates with a study conducted by Ahinkorah [38] who found that the likelihood of unintended pregnancy was high among women who lived with their partners. Similar observation was reported by Ameyaw et al., [3] who also reported that married women were 6 times more likely to report unintended pregnancy as compared to women who had never been married. A possible explanation for this finding is that women who live with their partners or are married may have more frequent sexual exposure which increases the likelihood of conception, particularly when effective contraceptive use is inconsistent or absent. In some relationships, limited negotiation power and partner influence over contraceptive decisions may also reduce women's ability to plan or prevent pregnancies. In contrast, a study conducted by Sarder et al., [39] found that women living with their partner were rather less likely to report unintended pregnancies. Similarly, research by Admasu et al., [40] revealed that single women and those who live away from their husbands had higher odds of reporting unintended pregnancies as compared to their counterparts. A possible explanation for this finding may be that single women and those living apart from their husbands may have less consistent access to contraceptive methods and weaker partner communication regarding fertility intentions. In many cases, sexual encounters among such women may be sporadic or unplanned, increasing the likelihood of unprotected intercourse. Moreover, social stigma surrounding contraceptive use among unmarried women often discourages them from seeking family planning services [41]. The absence of regular spousal support or joint decision-making on contraception may further limit their ability to prevent unintended pregnancies [42].

Furthermore, exposure to the internet was associated with lower odds of reporting unintended pregnancy, as reported in the adjusted model of the current study. While the crude analysis indicated a higher prevalence of unintended pregnancy among women who used the internet, this association is likely explained by the other covariates included that typically have greater internet access but may also exhibit higher baseline risk of unintended pregnancy. After controlling for these covariates, the adjusted model revealed a protective association which suggests that internet exposure may independently contribute to improved reproductive health awareness and safer decision-making. Supporting evidence include that of Kusumawardani et al., [43] which showed that premarital sexual activity can be influenced by exposure to online reproductive health resources. These resources may have one way or the other informed women to make better decisions as a result of greater knowledge and awareness regarding the consequences of premarital sexual activity. Another study by Sana [44] emphasised that online platforms provide accessible, evidence-based information on contraceptive methods, sexual health, and reproductive rights, which is crucial for informed decision-making. Possibly, exposure to the internet, a fast-growing communication channel such as TikTok, Snap Chat, Twitter, Instagram and Facebook could have exposed these women to the dangers of unintended pregnancies and safe sex practice measures and methods [45,46]. Moreover, the internet enables access to national campaigns on sex education such as the "You Only Live Once (YOLO)" movies and contents, partnered by the Health Promotion Department of the Ghana Health Service and other youth-led digital initiatives promoting safe sex, contraceptive literacy, and reproductive rights. This digital exposure may empower women to negotiate contraceptive use, recognize early signs of pregnancy, or adopt safer sexual behaviours, thereby limiting the likelihood of unintended pregnancy.

## Implications for policy and practice

The findings of this study have significant implications for reproductive health programming and policy in Ghana. The Regional Health Management Teams could pick clues from this study and intensify public health education campaigns that emphasize the consequences of unintended pregnancies and promote safe sex practices. These campaigns should be tailored to reach specific high-risk groups, particularly women over the age of 20 and those living with partners or spouses, who were shown to have higher odds of reporting unintended pregnancies. Moreover, the Ministry of Health in collaboration with the Ghana Health Service, could also leverage digital platforms to disseminate reproductive health information. Given the study's finding that internet access has a protective effect against unintended pregnancies, integrating online education into existing sexual and reproductive health strategies may enhance outreach and engagement, especially among younger and digitally active populations.

Furthermore, policymakers should prioritize the expansion of free contraceptive services in underserved and high-risk areas, particularly in rural parts of the Savannah region. This includes ensuring consistent supply chains, culturally sensitive service delivery, and community engagement to reduce stigma and misinformation around contraceptive use. Lastly, future research should adopt mixed-methods approaches to capture the nuanced socio-cultural, behavioral, and structural factors influencing unintended pregnancies. This will provide deeper insight into the underlying causes and inform more context-specific interventions while helping to minimize response bias in studies addressing sensitive topics. Reducing the burden of unintended pregnancies will not only improve maternal and child health outcomes but also advance gender equality, women's empowerment, and health system efficiency. These efforts directly support the achievement of Sustainable Development Goal 3 (Good Health and Well-being) and Goal 5 (Gender Equality), particularly targets 3.7 and 5.6, which emphasize universal access to sexual and reproductive health services and rights.

### Strengths and limitations

While this study benefits from the use of nationally representative data and rigorous statistical analysis, it has some limitations which must be acknowledged. The cross-sectional design limits the ability to draw causal conclusions, as it captures information at only one point in time. This means it cannot determine whether unintended pregnancies are a result of specific risk factors or merely associated with them. Additionally, self-reported data may lead to under-reporting of unintended pregnancies due to social stigma. It is important to note that some subgroups (e.g., women aged 45–49 and those whose first sexual intercourse occurred at ≥25 years) contained very few respondents, resulting in wide confidence intervals and reduced stability of the associated estimates. Although we retained these categories to maintain alignment with DHS coding and comparability with existing literature, the findings should be interpreted cautiously. The dataset also lacked important contextual variables such as partner attitudes and quality of care, which may have influenced the outcomes. Future research should consider using mixed-methods approaches to better understand the unusually high rates of unintended pregnancy, particularly in the Savannah region.

### Conclusions

The study concludes that more than half of the pregnancies among sexually active women in Ghana are unplanned, which is a very high rate. Living in the Savannah region, older age, living with a spouse, having low levels of education, and having access to the internet were factors reported to influence unintended pregnancy in this study. These findings emphasize the urgent need for targeted, equity-focused interventions to improve access to modern contraceptives, enhance comprehensive sexuality education, and address socio-cultural barriers to family planning.

### Acknowledgments

The authors express their gratitude to the DHS program for providing them access to the 2017 Ghana Maternal Health Survey database, as well as the survey respondents for their participation.

### Author contributions

**Conceptualization:** Samuel Salu.

**Data curation:** Samuel Salu.

**Formal analysis:** Samuel Salu.

**Writing – original draft:** Samuel Salu, Doreen Selasie Tay, Clinton Sekyere Frempong, Dennis Kweku Mawuli Okyere, Prince Tsekpetse.

**Writing – review & editing:** Samuel Salu.

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
