## [Decision Letter · Decision Letter 0]

20 Oct 2025

PGPH-D-25-01943

Why are so many pregnancies still unintended in Ghana? a closer look at factors influencing reproductive autonomy

Dear Dr. Salu,

Thank you for submitting your manuscript to PLOS Global Public Health. After careful consideration, we feel that it has merit but does not fully meet PLOS Global Public Health’s publication criteria as it currently stands. Therefore, we invite you to submit a revised version of the manuscript that addresses the points raised during the review process.

Please note that we have only been able to secure a single reviewer to assess your manuscript. We are issuing a decision on your manuscript at this point to prevent further delays in the evaluation of your manuscript. Please be aware that the editor who handles your revised manuscript might find it necessary to invite additional reviewers to assess this work once the revised manuscript is submitted. However, we will aim to proceed on the basis of this single review if possible.

Please revise your manuscript to address the reviewer's comments. Please note that while scientific rigor rather than novelty is our focus, we do ask that authors clearly specify where their manuscript fits within the literature. Please provide a point-by-point response to the reviewers upon resubmission.

We look forward to receiving your revised manuscript.

Kind regards,

Sarah Jose, Ph.D.

Staff Editor

Journal Requirements:

1. Please include a separate legend or caption for Figure 1 in your manuscript.

Additional Editor Comments (if provided):

Reviewers' comments:

Reviewer's Responses to Questions

**Comments to the Author**

1. Does this manuscript meet PLOS Global Public Health’s publication criteria?

Reviewer #1: Yes

2. Has the statistical analysis been performed appropriately and rigorously?

Reviewer #1: Yes

3. Have the authors made all data underlying the findings in their manuscript fully available (please refer to the Data Availability Statement at the start of the manuscript PDF file)?

Reviewer #1: Yes

4. Is the manuscript presented in an intelligible fashion and written in standard English?

Reviewer #1: Yes

Reviewer #1: Overall comment

This is a well-written paper that uses a nationally representative dataset to examine unintended pregnancy among sexually active women in Ghana. The topic is important and the statistical methods are sound. However, the novelty claim is not fully convincing because many of the identified predictors are consistent with prior research among the general reproductive-age population. The authors could strengthen their manuscript by more clearly articulating how their subgroup focus yields new insights, by comparing effect sizes and patterns with broader-population studies, and by exploring potential interaction effects. Below are section-specific comments.

Introduction

The research gap presented in lines 75–81 is weak and does not convincingly justify the novelty of focusing on sexually active women. You can make this section stronger by incorporating responses to the following:

1. Can you cite specific Ghanaian or sub-Saharan African studies that have aggregated sexually active women with the broader reproductive-age group and explain how this might have masked important differences?

2. Are there studies from other contexts that have shown different determinants for sexually active women compared to the broader group? If yes, why do you expect similar or different patterns in Ghana?

3. How much might the prevalence of unintended pregnancy be underestimated when all reproductive-age women (including those not sexually active) are analyzed together?

4. What specific new insights do you expect this subgroup analysis to reveal that could not be captured by previous broad-population studies?

5. How would identifying determinants in this subgroup change the design or targeting of Ghana’s reproductive health programs?

6. What specific new insights do you expect this subgroup analysis to reveal that could not be captured by previous broad-population studies?

7. How would identifying determinants in this subgroup change the design or targeting of Ghana’s reproductive health programs?

8. Why is the 2017 GMHS particularly suitable for answering this question now, does it contain variables or a sample structure that earlier datasets lacked?

9. How exactly are you defining this group in the context of your analysis (e.g., past month, past three months, past year), and why is that definition important for understanding unintended pregnancy risk?

10. How does your focus on sexually active women intersect with other high-risk groups, such as adolescents, unmarried women, or rural populations, and why not focus on those instead?

Methods

1. Please clarify the operational definition of “sexually active women” in the GMHS, including the recall period, and discuss potential implications for the prevalence estimate.

2. The exclusion criteria (missing responses, “don’t know” answers, and those never sexually active) removed a large proportion of the original sample; please assess and discuss the potential for selection bias.

3. The inclusion of independent variables only if they were significant at p≤0.05 in bivariate analysis may omit important confounders; consider including variables based on theoretical relevance as well.

4. Confirm that the complex survey design and clustering were fully accounted for in both descriptive and regression analyses.

5. Please explain the rationale for not including variables such as contraceptive use, marital status type, or partner’s fertility preferences, which are known predictors of unintended pregnancy.

Results

1. Many associations (age, education, region, cohabitation) are well-established in the literature. Please highlight any differences in magnitude or direction when compared to findings for the broader reproductive-age group in Ghana or SSA.

2. The high prevalence in the Savannah region (86.3%) is striking; please provide sample size and confidence intervals to assess precision.

3. Internet use shows an interesting reversal from crude to adjusted associations; please explore possible confounding or suppression effects.

**Do you want your identity to be public for this peer review?** For information about this choice, including consent withdrawal, please see our Privacy Policy

Reviewer #1: No

---

## [Decision Letter · Decision Letter 1]

3 Dec 2025

PGPH-D-25-01943R1

Why are so many pregnancies still unintended in Ghana? a closer look at factors influencing reproductive autonomy

Dear Dr. Salu,

Thank you for submitting your manuscript to PLOS Global Public Health. After careful consideration, we feel that it has merit but does not fully meet PLOS Global Public Health’s publication criteria as it currently stands. Therefore, we invite you to submit a revised version of the manuscript that addresses the points raised during the review process.

We look forward to receiving your revised manuscript.

Kind regards,

Adriana Biney

Academic Editor

Journal Requirements:

Additional Editor Comments (if provided):

Reviewers' comments:

Reviewer's Responses to Questions

**Comments to the Author**

Reviewer #1: All comments have been addressed

Reviewer #2: (No Response)

publication criteria?

Reviewer #1: Yes

Reviewer #2: Yes

3. Has the statistical analysis been performed appropriately and rigorously?

Reviewer #1: Yes

Reviewer #2: Yes

4. Have the authors made all data underlying the findings in their manuscript fully available (please refer to the Data Availability Statement at the start of the manuscript PDF file)?

Reviewer #1: Yes

Reviewer #2: No

5. Is the manuscript presented in an intelligible fashion and written in standard English?

Reviewer #1: Yes

Reviewer #2: Yes

Reviewer #1: I am satisfied with how you have thoroughly addressed all my previous comments. This is a well-written manuscript that makes excellent use of a nationally representative dataset to examine unintended pregnancy among sexually active women in Ghana. The topic is important, timely, and relevant for informing public health policy and practice. The statistical methods are sound and appropriately applied.

The revisions have significantly strengthened the manuscript, and it is now in excellent shape. The paper fully meets PLOS Global Public Health’s publication criteria and is suitable for publication in its current form.

Reviewer #2: Reviewer Comments

Manuscript Title: Why are so many pregnancies still unintended in Ghana? A closer look at factors influencing reproductive autonomy

Journal: PLOS Global Public Health

1. Overall Assessment

This manuscript examines the prevalence and determinants of unintended pregnancy among sexually active women in Ghana using the 2017 Ghana Maternal Health Survey. Focusing exclusively on sexually active women is a notable strength and adds nuance to existing literature that often aggregates all women of reproductive age. The topic is timely, relevant, and important for public health, reproductive autonomy, and policy targeting. The manuscript is generally well-structured and addresses a significant gap by focusing on a subgroup at genuine biological risk of pregnancy. The statistical methods are mostly appropriate and well presented. However, to strengthen the scientific rigour and the interpretability of findings, several primary clarification and methodological issues must be addressed.

2. Comments

Study Design & Methods

The manuscript requires clarification regarding key methodological choices. First, excluding all “don’t know” responses and missing values resulted in the removal of approximately 16,000 respondents, leaving only 1,453 participants; this substantial reduction raises concerns about potential selection bias and warrants justification for whether these exclusions may have systematically affected prevalence estimates. Second, the outcome definition for unintended pregnancy relies solely on responses from women who have ever been pregnant, yet the denominator appears to include women who may have never been pregnant, which could underestimate the proper proportion of unintended pregnancies; therefore, the authors should clarify whether nulligravid but sexually active women were excluded from the analysis or specify how they were accounted for.

Data Analysis & Results

The manuscript would benefit from clearer methodological and presentation improvements. Some subgroups, such as women aged 45–49 or those whose first sexual intercourse occurred at age ≥25, produce vast confidence intervals due to very small sample sizes; the authors should either collapse these categories or acknowledge the resulting instability. The multicollinearity assessment also needs clarification. The reported VIF of 1.44 is unclear; please specify whether this represents the mean VIF. Additionally, the variable on internet use shows inconsistent patterns: higher unintended pregnancy in bivariate analyses but a protective association after adjustment. This discrepancy warrants further interpretation.

Interpretation & Discussion

The discussion section requires several refinements to ensure accuracy and depth of context. First, causal language should be avoided given the cross-sectional design; for example, statements implying that “internet use reduces unintended pregnancy” should instead state that internet use is “associated with lower odds.” Second, the manuscript relies heavily on comparisons with Ethiopia, Nigeria, Bangladesh, and Indonesia but does not adequately situate the findings within Ghana’s specific reproductive health context, including its policy environment, cultural norms, and health system structure. Third, the notably significant adjusted effect observed in the Savannah region (aOR ≈ 5.5) warrants a more thorough contextual explanation, potentially relating to factors such as nomadic or hard-to-reach populations and cultural norms. Finally, several claims, particularly those concerning sexual education, the influence of digital platforms like TikTok, Snapchat, and Instagram, and gaps in the health system, require proper citations to support them.

Novelty & Contribution

The study’s focus on sexually active women is clearly articulated and adds meaningful value. However, the novelty claim would be more compelling if the authors explicitly compared their findings with studies that analyze the whole reproductive-age population and demonstrated how limiting the sample to sexually active women affects prevalence estimates or effect sizes.

3. Recommendation

A major revision is needed due to several concerns, including methodological uncertainties related to sample exclusions, the definition of sexual activity, and the outcome denominator, as well as interpretation issues that may lead to overstatement. Despite these issues, the topic remains important, and the manuscript has strong potential once these problems are addressed.

**Do you want your identity to be public for this peer review?** For information about this choice, including consent withdrawal, please see our Privacy Policy

Reviewer #1: No

Reviewer #2: No

---

## [Decision Letter · Decision Letter 2]

30 Dec 2025

PGPH-D-25-01943R2

Why are so many pregnancies still unintended in Ghana? a closer look at factors influencing reproductive autonomy

Dear Dr. Salu,

Thank you for submitting your manuscript to PLOS Global Public Health. After careful consideration, we feel that it has merit but does not fully meet PLOS Global Public Health’s publication criteria as it currently stands. Therefore, we invite you to submit a revised version of the manuscript that addresses the points raised during the review process.

We look forward to receiving your revised manuscript.

Kind regards,

Adriana Biney

Academic Editor

Journal Requirements:

Additional Editor Comments :

Your paper has addressed all the reviewers' comments and has improved significantly. However, there is still clarity needed on how you attained your sample size since the sample reduced significantly. Readers need to understand who exactly were filtered out as missing and 'don't know'. I would encourage you to address the following comments prior to the manuscript being published.

Abstract

Line 30 – Change 'The study concluded' to 'The study concludes'.

Introduction

Line 70 – Revise as 'The extant literature has…'

Line 71 – Move the references (5,11) to the end of the sentence with the other references.

Lines 75-90 - These paragraphs provide a good indication of the gap, however, based on your small sample size, readers need to be sure about who exactly was selected for your sample. Is it limited to currently pregnant or ever pregnant women? If so, indicate the variable used to filter other women out. Then let the exact target population reflect in the paragraphs, in that not only does your study limit the sample to sexually active women, it limits it further to those currently/ever pregnant. Ensure that the specific population is discussed and this runs throughout the paper.

Methods

Lines 130-131 - again, provide clarity on which variables you used to exclude women due to their missing and don't know responses. If it is the currently or ever pregnant variable? If so, then indicate this. An unweighted frequency of the age at first sex variable indicates 3,665 had never had sex, 5 refused to answer and over 21,000 had ever had sex. If you are defining those sexually active as those who had sex in the past year or month, and thus filtered many women out, then this needs to be clear.

Line 152 - tell us which regions were combined in each of the three zones.

Results

Check values in Table 1 – used the internet in the last 12 months. The same value has been written twice.

Discussion

In the Discussion, the authors can also reflect on how unintended pregnancy is also higher when fertility intention declines among a group - when people want fewer children, they are more likely to declare additional pregnancies as unintended.

The authors should check other publications for the in-text citation format which usually includes the author’s last name and the reference – for instance 'Nyarko [24] stated that' or 'Sana [45] found that'.

Reviewers' comments:

Reviewer's Responses to Questions

**Comments to the Author**

Reviewer #2: All comments have been addressed

publication criteria?

Reviewer #2: Yes

3. Has the statistical analysis been performed appropriately and rigorously?

Reviewer #2: Yes

4. Have the authors made all data underlying the findings in their manuscript fully available (please refer to the Data Availability Statement at the start of the manuscript PDF file)?

Reviewer #2: Yes

5. Is the manuscript presented in an intelligible fashion and written in standard English?

Reviewer #2: Yes

Reviewer #2: SECOND-ROUND REVIEWER REPORT

The manuscript has improved substantially in clarity, contextualization, and justification of the analytical focus on sexually active women. Several of the original concerns have been adequately addressed.

The revised manuscript clearly justifies the restriction of the analysis to sexually active women, strengthening its overall contribution, and now transparently reports the scale and reasons for sample exclusions while appropriately acknowledging the potential for selection bias, even though additional supplementary comparisons could further enhance transparency. The authors have improved the interpretation of internet use by addressing discrepancies between crude and adjusted associations and supporting this Discussion with relevant literature. The Discussion is better contextualized within Ghana, incorporating policy, regional, and health system perspectives, particularly for the Savannah region, and the authors have avoided causal language while clearly articulating the study’s novelty relative to prior analyses of the general reproductive-age population.

Recommendation

The manuscript shows strong potential, has improved meaningfully, and can be considered for publication.

**Do you want your identity to be public for this peer review?** For information about this choice, including consent withdrawal, please see our Privacy Policy

Reviewer #2: No

---

## [Editor Report · Decision Letter 3]

5 Jan 2026

Why are so many pregnancies still unintended in Ghana? a closer look at factors influencing reproductive autonomy

PGPH-D-25-01943R3

Dear Mr Salu,

We are pleased to inform you that your manuscript 'Why are so many pregnancies still unintended in Ghana? a closer look at factors influencing reproductive autonomy' has been provisionally accepted for publication in PLOS Global Public Health.

Best regards,

Adriana Biney

Academic Editor